# Adolescent Use of Dating Applications and the Associations with Online Victimization and Psychological Distress

**DOI:** 10.3390/bs13110903

**Published:** 2023-11-02

**Authors:** Tzu-Fu Huang, Chun-Yin Hou, Fong-Ching Chang, Chiung-Hui Chiu, Ping-Hung Chen, Jeng-Tung Chiang, Nae-Fang Miao, Hung-Yi Chuang, Yen-Jung Chang, Hsi Chang, Hsueh-Chih Chen

**Affiliations:** 1Department of Health Promotion and Health Education, National Taiwan Normal University, Taipei 10610, Taiwan; gh52025@gmail.com (T.-F.H.); punny63@gmail.com (C.-Y.H.); yjchang@ntnu.edu.tw (Y.-J.C.); 2Department of Family Medicine, Taipei City Hospital, Zhongxiao Branch, Taipei 11556, Taiwan; 3Graduate Institute of Information and Computer Education, National Taiwan Normal University, Taipei 10610, Taiwan; cchui@ntnu.edu.tw; 4The Graduate Institute of Mass Communication, National Taiwan Normal University, Taipei 10610, Taiwan; pxc24@ntnu.edu.tw; 5Department of Statistics, National Chengchi University, Taipei 11605, Taiwan; chiangj@nccu.edu.tw; 6Post-Baccalaureate Program in Nursing, College of Nursing, Taipei Medical University, Taipei 11031, Taiwan; naefang@tmu.edu.tw; 7Department of Public Health, Kaohsiung Medical University, Kaohsiung 80708, Taiwan; ericch@kmu.edu.tw; 8Department of Pediatrics, School of Medicine, College of Medicine, Taipei Medical University, Taipei 11031, Taiwan; jamesc@h.tmu.edu.tw; 9Department of Educational Psychology and Counseling, College of Education, National Taiwan Normal University, Taipei 10610, Taiwan; chcjyh@ntnu.edu.tw

**Keywords:** adolescence, dating applications, online victimization, psychological distress

## Abstract

In this study, we examined the relationships between the use of online dating applications (apps), online victimization, and psychosocial distress among adolescents. This study was conducted in 2020. A sample of 2595 seventh-grade students from 30 Taiwanese middle schools was surveyed. We conducted a self-administered survey. Overall, 15% of the adolescents reported using online dating apps in the past year, while 78% reported having seen dating app advertisements on the internet in the past year. Multivariate analysis results indicated that adolescents’ exposure to the marketing of dating apps and poor academic performance were both associated with the use of online dating apps. Adolescents who used dating apps were more likely to experience online privacy victimization, cyberbullying victimization, and online sexual harassment. The use of dating apps by adolescents, online privacy victimization, cyberbullying victimization, and online sexual harassment were associated with higher levels of depression, anxiety, and stress. In conclusion, adolescent use of dating apps is related to online victimization and psychological distress.

## 1. Introduction

The use of dating applications (apps) by adolescents is an emerging public health issue. The use of dating apps has rapidly increased in the past 10 years. In 2022, there were more than 366 million online dating service users [1]. About 30% of adults in the United States (U.S.) have ever used online dating apps, while 35% of online dating users report being harassed or sent sexually explicit messages [2]. A study of Norwegian university students found that half had used mobile dating apps [3], while an Australian survey found that three-quarters of dating app users were subjected to online dating-app-facilitated sexual violence victimization [4]. Although most dating apps claim to require users to be at least 18 years of age, adolescents routinely report using dating apps. A cross-national survey of Finnish, American, Spanish, and South Korean students found that 15% of adolescents use dating apps at least sometimes, while online dating app use by adolescents has been associated with a greater likelihood of victimization via online harassment, online sexual harassment, and other cybercrimes and sexual victimization by adults as well as peers [5]. Prior studies that mainly canvassed young adults found that the use of dating apps poses risks associated with privacy, security, intimacy [6,7,8], sexting victimization [9], sexual assaults and sexual abuse [10,11], risky sexual behaviors [12,13,14], and sexually transmitted diseases [15,16,17].

Review studies have associated the use of online dating apps with an increase in online sexual victimization risks, and have referred to such apps as avenues of technology-facilitated sexual violence perpetration [18,19] and as places where technology-facilitated sexual violence victimization is a prevalent phenomenon [20]. According to lifestyle exposure theory, the use of dating apps by adolescents represents a risky online lifestyle, while unsafe activities in relationship formation and information sharing may make victimization more likely [5]. Studies have shown that the prevalence of cyber sexual victimization (e.g., online sexual harassment, cyberstalking, sexting) among adolescents ranges from 18 to 66% [21]. Approximately 28% of U.S. middle and high school students in a relationship had been the victim of digital dating abuse. In that study, students’ depressive symptoms, sexual intercourse, sexting, and being the victim of cyberbullying were associated with digital dating abuse [22]. Another study found that among young adults with dating experience, 76% had experienced digital dating abuse, while the average age of initiation for digital dating abuse behaviors was 16 years, with 11 years of age being the earliest [23]. In addition, the studies have positively associated adolescent online dating violence with physical, sexual, and verbal dating violence [24,25], and cyber dating abuse has been positively associated with traditional abuse [22,25].

Research has established that the corollaries of online dating app use involve online victimization and negative psychological impacts. Adolescents who use online dating apps engage in more online risky activities during online communication and information sharing [5], and engaging in sexting and online risk behaviors have been significantly linked to digital dating victimization [26,27]. In at least one study, approximately 30% of adolescents reported experiencing cyberbullying victimization, while 13% reported at least two incidences of cyberbullying–online grooming, and 7% had dealt with at least three incidences of cyberbullying–sexting–grooming [28]. In addition, studies have connected the use of online dating apps with increases in depression [29], distress [30], anxiety [31], low self-esteem [27], body dissatisfaction [32], and disordered eating [33], and excessive swiping was also found to be detrimental to the well-being of young dating app users [34]. Studies have found that parent–adolescent attachment and parenting practices could prevent online risky behaviors and dating victimization [35,36,37]. Parental mediation and parenting style play crucial roles in protecting adolescents from the dangers of cyberbullying victimization and online risks [38,39,40,41,42,43].

Taiwan is experiencing increases in the marketing of dating apps on the internet, social media, television, and outdoor billboards. A survey conducted in Taiwan found that more than one-third of children and adolescents reported having ever used dating apps and engaging in online and offline risks, such as providing personal information on the internet, having met people from the apps in person, and sending nude pictures of themselves [44]. There have been a number of news reports describing the sexual victimization of adolescents and crimes involving dating app use in the Taiwan media. Research indicates that sexual victimization is increasingly facilitated by technology, but the prevalence, dynamics, and prevention of such offenses have not been studied in depth [18,45]. Prior studies regarding dating app use have primarily investigated adults or men who have sex with men [6,46], while only scant amounts of research have assessed the extent, correlates, and impact of adolescent online dating app use. Adolescents, who are dealing with puberty, are more vulnerable to online dating app use. Thus, this study focused on the prevalence and correlates of dating app use among seventh-grade students in Taiwan and the relationships to online victimization and psychological distress.

## 2. Materials and Methods

### 2.1. Participants

This cross-sectional study used the probability-proportional-to-size method to collect data from thirty Taiwanese middle schools in 2020. However, about two-thirds of schools refused to participate in the survey. Seventh-grade students were recruited for the study. Trained interviewers conducted this study and utilized a self-administered questionnaire survey in classrooms. For the participant schools, we issued consent forms to both students and parents. We stipulated that student information would be protected and would remain anonymous. Of the students and parents who agreed to participate in this study, about 96% completed the survey. Overall, a total of 2595 seventh-grade students completed the questionnaire.

### 2.2. Instrument

The questionnaire was developed on the basis of previous research. The content validity of the questionnaire was assessed by six experts. The questionnaires were distributed to the students by the interviewers in the classroom. Moreover, the questionnaire scales were pre-tested to evaluate their reliability. National Taiwan Normal University’s Institutional Review Board approved this study (201905HS068). We have provided informed consent to the students and described details that included the aims, methods, procedures, expectations, and participants’ rights in this study.

### 2.3. Material and Measures

#### 2.3.1. Dating App Use

Dating app use was measured through one item. The students were asked, “How often do you download and use dating apps?”. The response options were as follows: 1, never; 2, ever before a year; 3, a few times within a year; 4, a few times within a month; and, 5, a few times within a week. If the students answered “a few times within a year” or more frequently, they were categorized as having used dating apps.

#### 2.3.2. Online Privacy Victimization

Online privacy victimization was measured using two items. The students were asked “How often has your online account or password been stolen?”, and “How often has your personal information been shared online without your permission?”. The response options were as follows: 1, never; 2, ever before a year; 3, a few times within a year; 4, a few times within a month; and, 5, a few times within a week. If the students answered “a few times within a year” or more frequently, for either item, they were categorized as having experienced online privacy victimization.

#### 2.3.3. Cyberbullying Victimization

Cyberbullying victimization was measured with their answers to four questions: How often has someone made rude comments about you online?; How often has someone spread rumors about you online to hurt you maliciously?; How often has someone sent you nude pictures or videos online to harass you?; and, How often have you been cyberbullied by classmates or friends? The response options were as follows: 1, never; 2, ever before a year; 3, a few times within a year; 4, a few times within a month; and, 5, a few times within a week. If the students answered “a few times within a year” or more frequently for any of the four items, they were categorized as having experienced cyberbullying victimization.

#### 2.3.4. Online Sexual Harassment

Online sexual harassment was measured with the answers to four questions that were adapted from the U.S. Youth Internet Safety Surveys [47,48]: How often has someone been threatened to post your private pictures or information online?; How often has someone threatened to post a sexual picture or video of you online?; How often has someone sent you unwanted sexual messages online?; and, How often has someone asked you to do something sexual online that you did not want to?. The response options were as follows: 1, never; 2, ever before a year; 3, a few times within a year; 4, a few times within a month; and, 5, a few times within a week. If the students answered “3. a few times within a year” or more frequently for any of the four items, they were categorized as having experienced online sexual harassment.

#### 2.3.5. Psychological Distress

Psychological distress was measured with the Chinese version of Depression, Anxiety, Stress Scale (DASS-21) [49]. The DASS-21 consists of 21 items including depression (7 items), anxiety (7 items), and stress (7 items). The response options for each item were rated on a 4-point scale ranging from 0 (did not apply to me at all) to 3 (applied to me very much, or most of the time). Higher scores indicated higher levels of depression, anxiety, and stress.

#### 2.3.6. Dating App Marketing Exposure

Dating app marketing exposure was measured using three items: During the past year, how often have you seen dating app advertisements online?; During the past year, how often have you seen dating app advertisements in retail stores or on outdoor billboards?; and, During the past year, how often have you seen influencers promoting dating apps?. The response options were on a 5-point scale ranging from 1 (never) to 5 (always). The higher the scores, the more exposure to marketing on dating apps.

#### 2.3.7. Parental Mediation

The parental mediation scales were adapted from a previous study [50] and consisted of 20 items. The students were asked about their experience with three types of parental mediation: parental restrictive mediation (7 items), parental active mediation (8 items), and parental monitoring mediation (5 items). A sample question about parental restrictive mediation is “Do your parents set rules about how long you are allowed to go online?”. A sample question about parental active mediation is “Do your parents talk to you about how to use the internet safely?”. A sample question about parental monitoring mediation is “Do your parents check your browsing history?”. The response options were on a 5-point scale ranging from 1 (never) to 5 (always). Higher scores indicate greater levels of parental mediation.

#### 2.3.8. Adolescent Characteristics

Adolescent characteristics canvassed in the present study include gender (male (coded 1), (female (coded 0)); area (rural (coded 1); urban (coded 0)); academic performance (below average (coded 1), average or above (coded 0)); and, household income (lower income class or lower-middle income class (coded 1); median or upper-income class (coded 0)).

### 2.4. Statistical Analysis

This study used SAS version 9.4 for Windows to perform the statistical analyses. Descriptive statistics such as percentages and means were performed for all the variables. T-tests and Chi-squared tests were used to examine gender differences in dating app marketing exposure, dating app use, online victimization, and parental mediation. A series of logistic regression models were conducted to examine the factors related to dating app use and online victimization. In addition, multiple regression modeling was used to examine factors associated with depression, anxiety, and stress.

## 3. Results

### 3.1. Adolescents’ Sociodemographic Characteristics

Among the 2595 seventh-grade student participants, 58.9% were from urban areas, while 41.1% came from rural regions. One-fifth of the students came from households experiencing poverty, and 28% of the students reported below-average academic performance.

### 3.2. Adolescents’ Dating App Use and Online Victimization Experiences

Overall, 15% of the adolescents reported having used dating apps in the past year. In addition, 6% of the adolescents had experienced online privacy risks in the past year such as stolen passwords (5%). A total of 13% of the adolescents had experienced cyberbullying such as rude comments in the past year (8%), and 4% of adolescents had experienced online sexual harassment, such as the sending of unwanted messages (3%). By gender, girls had higher rates (6%) of online sexual harassment experiences compared with boys (2%) in the past year, while boys had higher rates (7%) of online privacy risks than those experienced by girls (5%) (Table 1).

### 3.3. Adolescents’ Exposure to Dating App Marketing

Overall, 78% of the adolescents reported that they had seen dating app advertisements on the internet in the past year, while 65% had seen influencer marketing of dating apps, and 46% had seen dating app advertisements in retail stores or on outdoor billboards. The mean score of exposure to dating app marketing was 2.30, while the girls reported higher levels of dating app marketing exposure compared with that reported by boys (Table 2).

### 3.4. Adolescents’ Psychological Distress and Parental Mediation

The Cronbach’s alpha values of depression, anxiety, and stress in the present study were 0.89, 0.80, and 0.88, respectively. The Cronbach’s alpha values of parental mediation in the present study were 0.81 (parental restrictive mediation), 0.93 (parental active mediation), and 0.83 (parental monitoring mediation). Overall, the mean scores of depression, anxiety, and stress were 5.72, 4.73, and 7.20, respectively, while the girls had higher scores of depression, anxiety, and stress compared with those experienced by the boys. In addition, the adolescents reported that the mean scores of parental restrictive mediation, parental active mediation, and parental monitoring mediation were 2.54, 2.61, and 1.74, respectively, while the girls reported greater levels of parental active mediation compared with the boys (Table 2).

### 3.5. Factors Related to Dating App Use

The simple logistic regression analysis results indicate that factors related to adolescents’ dating app use include the following: poor academic performance, household poverty, dating app marketing exposure, lower levels of parental restrictive mediation, and lower levels of parental active mediation. The multivariate logistic regression analysis results indicated that adolescents who had poor academic performance and had higher levels of dating app marketing exposure were more likely to use dating apps (Table 3). 

### 3.6. Relationships between Dating App Use and Online Victimization

The multiple logistic regression analysis results indicate that boys who have poor academic performance and use dating apps are more likely to experience online privacy risks. In addition, adolescents living in household poverty, with lower levels of parental restrictive mediation, higher levels of parental active mediation, and who use dating apps are more likely to experience cyberbullying. Girls who use dating apps are more likely to experience online sexual harassment (Table 4).

### 3.7. Relationships among Dating App Use, Online Victimization, and Psychological Distress

Multiple regression analysis results indicate that girls with poor academic performance, household poverty, habitual use of dating apps, experiences with online privacy risks, and who have been cyberbullied and dealt with online sexual harassment are more likely to have higher rates of depression. In a similar manner, girls who live in household poverty, use dating apps, experience online privacy risks, have been cyberbullied, and have dealt with online sexual harassment are more likely to have higher rates of anxiety. Moreover, females who use dating apps, have experienced online privacy risks, have been cyberbullied, and have been victims of online sexual harassment also are more likely to have higher rates of stress (Table 5).

## 4. Discussion

In this study, 15% of adolescents had used online dating apps in the past year, which has been positively associated with online privacy risks, cyberbullying victimization, and online sexual harassment. These findings are consistent with prior studies indicating that online dating exposes youth to various interpersonal risks [5,45]. A developmental stage where adolescents are eager to confirm their self-worth and expand their social connections makes dating apps an appealing platform. Since dating apps are designed to facilitate the free disclosure of personal information and intimate images [19], studies have shown that dating app users are more likely to encounter privacy risks [8], sexting victimization [9], and sexual abuse [10].

The results of this study show that 13% of adolescents have been cyberbullied, 6% have experienced online privacy risk, and 4% have endured online sexual harassment. Adolescent use of online dating and experience with online victimization have been positively associated with negative psychological consequences such as depression, anxiety, and stress. Studies have also shown that online dating app users tend to have higher levels of distress, anxiety, and depression [29,30,31]. Prior research has linked dark personality traits with reasons for using dating apps [51]. Since the different forms of online victimization that adolescents experience are related, preventive programs in schools and communities should be holistic [22,52].

Moreover, this study shows that about four-fifths of adolescents have seen dating app advertisements on the internet, while two-thirds of adolescents have seen influencer marketing of dating apps. Adolescents’ exposure to dating app marketing has been positively associated with dating app use. Although very little research has explored the impact that dating app marketing exerts on minors, a growing body of research has shown that increasingly sophisticated online marketing techniques are being used to promote the use of unhealthy commodities such as unhealthy food, beverages, alcohol, and electronic cigarettes among children and adolescents [53,54]. A UK study showed that governments are underestimating the impact of digital unhealthy food advertising restrictions [55]. The World Health Organization has advocated that global policy and international cooperation are needed to monitor and regulate cross-border digital marketing of unhealthy products to minors [56].

In the present study, a multivariate analysis model showed that the students whose parents more restrictively mediate their internet use were less likely to be victimized, while students whose parents implement a higher level of active mediation were associated with greater levels of cyberbullying victimization. A longitudinal study demonstrated that the effectiveness of active and restrictive mediation in relation to students’ cyberbullying experiences varies, particularly in the context of gender [39]. Some studies have shown that adolescents with parental restrictive mediation were less likely to experience cyberbullying victimization [40,41], online harassment [42], and online risks [43]. In another study, parental monitoring decreased risk, but the influence was effective mostly during early adolescence, which suggests the need to develop age-appropriate electronic dating violence prevention strategies [57].

In addition, the results of this study show that girls are more likely to experience online sexual harassment, with the attendant negative psychological impact. Prior studies have also shown that girls are more likely to be victims of technology-facilitated sexual violence [52]. Compared with boys, girls express a greater number of negative emotional responses to digital dating abuse victimization, which suggests that digital dating abuse is particularly detrimental for girls [58]. Moreover, the results of the present study show that adolescents with poor academic performance are more likely to use online dating apps, which increases their exposure to online privacy victimization and depression. Prior studies have shown that girls who did not fit in well at school and who had difficulty making friends are more likely to initiate romantic relationships online [59]. Another study showed that social anxiety is a factor that encourages the use of online dating apps [31]. It remains unclear, however, if it is the case that dating apps cause anxiety or if it is a matter of anxious individuals simply being more likely to use dating apps. There could be a reciprocal causality between online dating app use and anxiety.

The present study had some limitations. First, this was a cross-sectional study and could not demonstrate the causality of factors such as dating app use, online victimization, and the psychological impact on adolescents. Future longitudinal studies could be conducted to examine the causal relationships between dating app use, online victimization, and psychological distress. Second, self-reporting of dating app marketing exposure can be biased due to potential recall bias. Third, the measurement of gender was based on biological sex, while future studies could expand to include gender identity. Finally, social desirability bias may influence the truthfulness of adolescents’ reports of dating app use and online victimization experiences such as online sexual harassment. These factors could have led to an underestimation of the prevalence of dating app use and online victimization. However, confidentiality was emphasized in this study. Despite these limitations, the strength of this study was the large sample size that was used to examine the relationships among adolescents’ exposure to dating app marketing, online dating app use, online victimization, and psychological distress.

## 5. Implications

In accordance with the Lifestyle Exposure Theory, the present study positively associated adolescent use of dating apps with susceptibility to online victimization. This underscores the urgency of instituting comprehensive regulations to protect children and adolescents from the perils of exposure to dating app marketing. Furthermore, we advocate for the introduction of parental mediation training programs. These programs can empower parents to employ effective strategies that reduce online risks for their children. Additionally, there is a pressing need to incorporate media literacy and cyber dating violence prevention programs into the educational system, which could equip students with the essential skills needed to resist digital marketing and prevent cyber victimization.

## 6. Conclusions

This study explored the prevalence and correlates of adolescents’ dating app use and examined the relationships between online dating app use, online victimization, and psychosocial distress among adolescents. The results showed that 15% of the adolescents canvassed for this study reported using online dating apps in the past year, while more than three-quarters of the students reported seeing dating app advertisements on the internet. The multivariate analysis results positively associated adolescent exposure to dating app marketing and poor academic performance with online dating app use. The adolescents who used dating apps were more likely to experience online victimization such as online privacy victimization, cyberbullying victimization, and online sexual harassment. Adolescent dating app use and online victimization experiences were positively associated with depression, anxiety, and stress.

## Figures and Tables

**Table 1 behavsci-13-00903-t001:** Dating app use and online victimization risks by gender.

Variable	Overall		Girls		Boys		*p* Value
	*n*	%	*n*	%	*n*	%	
Area							0.0007
Urban	1529	58.9	849	62.0	680	55.5	
Rural	1066	41.1	520	38.0	546	44.5	
Academic performance							0.0066
Average or above	1780	71.7	971	74.01	809	69.1	
Below average	701	28.3	340	25.9	361	30.9	
Household poverty							0.0619
Yes	492	19.3	241	17.88	251	20.8	
No	2063	80.7	1107	82.12	956	79.2	
Dating apps use							0.3217
Yes	388	15.0	214	15.64	174	14.3	
No	2201	85.0	1154	84.36	1047	85.8	
Online privacy risk							0.0071
Yes	156	6.0	66	4.83	90	7.4	
No	2434	94.0	1300	95.17	1134	92.6	
Cyberbullying risk							0.9838
Yes	335	12.9	177	12.97	158	12.9	
No	2251	87.1	1188	87.03	1063	87.1	
Online sexual harassment							<0.0001
Yes	103	4.0	79	5.79	24	2.0	
No	2485	96.0	1286	94.21	1199	98.0	

Chi-square tests conducted. Overall *n* = 2595; girls *n* = 1369; boys *n* = 1226.

**Table 2 behavsci-13-00903-t002:** Dating app marketing exposure, parental mediation, and psychological distress by gender.

Variable	Overall		Girls		Boys		*p* Value
	Mean	SD	Mean	SD	Mean	SD	
Dating app marketing exposure	2.30	1.03	2.38	0.99	2.22	1.06	<0.0001
Parental restrictive mediation	2.54	0.98	2.56	0.98	2.53	0.99	0.456
Parental active mediation	2.61	1.18	2.72	1.15	2.50	1.20	<0.0001
Parental monitoring mediation	1.74	0.88	1.75	0.86	1.73	0.91	0.623
Depression score	5.72	8.26	6.20	8.57	5.18	7.87	0.002
Anxiety score	4.73	6.28	5.04	6.47	4.38	6.03	0.007
Stress score	7.20	8.63	8.01	9.01	6.29	8.10	<0.0001

T-tests conducted. Overall *n* = 2595; girls *n* = 1369; boys *n* = 1226.

**Table 3 behavsci-13-00903-t003:** Factors related to dating app use.

Variable	Simple Logistic Regression		Multivariate Logistic Regression	
	OR	95% CI	OR	95% CI
Area (rural vs. urban)	1.27	1.02–1.58	1.35	1.07–1.71
Gender (male vs. female)	0.90	0.72–1.11	0.94	0.74–1.18
Poor academic performance (yes vs. no)	1.37	1.09–1.74	1.31	1.02–1.68
Household poverty (yes vs. no)	1.46	1.13–1.89	1.32	1.00–1.73
Dating app marketing exposure	1.52	1.38–1.69	1.52	1.37–1.70
Parental restrictive mediation	0.86	0.77–0.96	0.98	0.83–1.16
Parental active mediation	0.90	0.82–0.99	0.94	0.83–1.07
Parental monitoring mediation	0.94	0.83–1.07	1.01	0.85–1.19

Multiple logistic regression conducted. Multivariate logistic regression model *n* = 2355.

**Table 4 behavsci-13-00903-t004:** Relationships between dating app use and online victimization risks.

Variable	Online Privacy Victimization		Cyberbullying Victimization		Online Sexual Harassment	
	OR	95% CI	OR	95% CI	OR	95% CI
Area (rural vs. urban)	0.73	0.51–1.05	0.91	0.71–1.17	0.87	0.56–1.36
Gender (male vs. female)	1.44	1.02–2.03	1.07	0.84–1.37	0.29	0.18–0.49
Poor academic performance (yes vs. no)	1.47	1.03–2.11	1.00	0.76–1.31	0.87	0.52–1.43
Household poverty (yes vs. no)	1.17	0.78–1.75	1.36	1.01–1.82	1.21	0.72–2.05
Parental restrictive mediation	0.82	0.64–1.05	0.75	0.63–0.89	0.91	0.68–1.22
Parental active mediation	0.93	0.76–1.13	1.19	1.04–1.36	1.05	0.84–1.33
Parental monitoring mediation	0.95	0.72–1.24	1.02	0.85–1.21	1.00	0.73–1.37
Dating app use	2.47	1.69–3.61	1.81	1.35–2.43	2.05	1.26–3.34

Multiple logistic regression conducted. Online privacy victimization model *n* = 2383. Cyberbullying victimization model *n* = 2381. Online sexual harassment model *n* = 2381.

**Table 5 behavsci-13-00903-t005:** Relationships between dating app use, online victimization risks, and psychological distress.

Variable	Depression			Anxiety			Stress		
	Β	SD	*p*	β	SD	*p*	β	SD	*p*
Intercept	4.49	0.28	<0.0001	4.10	0.22	<0.0001	6.76	0.30	<0.0001
Area (rural vs. urban)	−0.19	0.33	0.567	−0.04	0.25	0.885	0.15	0.35	0.668
Gender (male vs. female)	−0.84	0.32	0.010	−0.64	0.25	0.011	−1.67	0.34	<0.0001
Poor academic performance (yes vs. no)	0.98	0.37	0.007	0.29	0.28	0.297	−0.25	0.39	0.512
Household poverty (yes vs. no)	0.94	0.41	0.023	0.81	0.32	0.012	0.60	0.44	0.173
Dating app use	2.94	0.45	<0.0001	1.46	0.35	<0.0001	2.37	0.48	<0.0001
Online privacy risk	1.73	0.69	0.012	1.69	0.53	0.002	2.08	0.73	0.005
Cyberbullying victimization	3.98	0.51	<0.0001	2.44	0.39	<0.0001	4.22	0.54	<0.0001
Online sexual harassment	4.27	0.88	<0.0001	2.03	0.68	0.003	3.22	0.93	0.001

Multiple regression conducted. Depression model *n* = 2384, R^2^ = 8.78%. Anxiety model *n* = 2398, R^2^ = 5.21%. Stress model *n* = 2398, R^2^ = 7.26%.

## Data Availability

The data presented in this study can be requested and provided.

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
