# Peer review of "Adolescent Use of Dating Applications and the Associations with Online Victimization and Psychological Distress"

_behavsci, 2023, doi:10.3390/bs13110903_

Round 1

Reviewer 1 Report

Comments and Suggestions for Authors

The theme presented by the authors is pertinent, current and necessary, from an academic and scientific point of view. It also reveals relevant public impact.

Although the introduction is well structured and presents relevant data, it would be important to explore the essence of the article in more depth, creating a theoretical foundation that would allow the object of study to be framed in the scientific literature.

Regarding the methodological part, particularly with regard to the objectives and design of the research instrument, the article presents relevant and explanatory content. The design of the instrument is very detailed.

Regarding the characterization of the sample, this is insufficient. Who are the participants? It will be important to create a sociodemographic portrait of the sample, especially in order to avoid objectifying the participants.

The discussion is well structured and enriches the presented article. I would advise creating a final topic, possibly after the conclusion, about the implications of the results presented, for the present and future context of young people's lives, presenting some recommendations and research paths.

The bibliography presented is current and relevant.

Author Response

Thank you very much for taking the time to review this manuscript. Please find the detailed responses below and the revisions/corrections highlighted/in track changes in the re-submitted files.

Comments 1:

The theme presented by the authors is pertinent, current and necessary, from an academic and scientific point of view. It also reveals relevant public impact.

Although the introduction is well structured and presents relevant data, it would be important to explore the essence of the article in more depth, creating a theoretical foundation that would allow the object of study to be framed in the scientific literature.

Response 1: Thank you for your suggestion. Lifestyle exposure theory was added in the Introduction.

Comments 2:

Regarding the methodological part, particularly with regard to the objectives and design of the research instrument, the article presents relevant and explanatory content. The design of the instrument is very detailed.

Regarding the characterization of the sample, this is insufficient. Who are the participants? It will be important to create a sociodemographic portrait of the sample, especially in order to avoid objectifying the participants.

Response 2: Thank you for your suggestion. Adolescents’ sociodemographic characteristics were added to the Result section.

Comments 3:

The discussion is well structured and enriches the presented article. I would advise creating a final topic, possibly after the conclusion, about the implications of the results presented, for the present and future context of young people's lives, presenting some recommendations and research paths.

Response 3: Thank you for your suggestion. The implications of the results were added to the Discussion section.

Comments 4:

The bibliography presented is current and relevant.

Response 4: Thank you for your positive affirmation.

We are grateful for your help.

Reviewer 2 Report

Comments and Suggestions for Authors

This study is not a reliable one since 96% turnover from the respondents receiving the survey is not realistic and can not be acceptable as a legitimate source to base the study on. Another concern is that the gender distinction has been made on a binary scope (male/female) whereas the scope must be wider to cover gender identity of all participants.

Comments on the Quality of English Language

not very problematic.

Author Response

Thank you very much for taking the time to review this manuscript. Please find the detailed responses below and the revisions/corrections highlighted/in track changes in the re-submitted files.

Comments 1:

This study is not a reliable one since 96% turnover from the respondents receiving the survey is not realistic and can not be acceptable as a legitimate source to base the study on. Another concern is that the gender distinction has been made on a binary scope (male/female) whereas the scope must be wider to cover gender identity of all participants.

Response 1: We do apologize for the misunderstanding. We tried to report the number of participating schools and students more clearly. We added the description in the participant section.

Thank you for the suggestion. We collected information based on biological sex. In the Discussion section, we have incorporated the recommendation to explore gender identity in future studies.

Comments 2:

Comments on the Quality of English Language not very problematic.

Response 2: Thank you for your help.

Reviewer 3 Report

Comments and Suggestions for Authors

The manuscript is well written and there are minor improvements that I can suggest:

In the Methodology part: I can suggest that the Cronbach's alpha test results to be moved to the results.

In the Results part: please take away from the first paragraphs words like approximatively or about from before the percentage number. From a visualization standpoint please intercalate the paragraphs with the text as to have the table closer to the paragraph that is cited in.

Discussions: you have a good introduction, but still there are new citations added in the discussion chapter. Please make sure that you do not have new citations in the discussion section, hence improve the introduction so that you give a better status quo to which you compare your results with.

Author Response

Thank you very much for taking the time to review this manuscript. Please find the detailed responses below and the revisions/corrections highlighted/in track changes in the re-submitted files.

Comments 1:

The manuscript is well written and there are minor improvements that I can suggest:

In the Methodology part: I can suggest that the Cronbach's alpha test results to be moved to the results.

Response 1: Thank you for your suggestion. We have moved the Cronbach's alpha test results to the Result section.

Comments 2:

In the Results part: please take away from the first paragraphs words like approximatively or about from before the percentage number. From a visualization standpoint please intercalate the paragraphs with the text as to have the table closer to the paragraph that is cited in.

Response 2: Thank you for your suggestion. We have taken away from the first paragraph words like approximatively or about from before the percentage number. The table was closer to the paragraph that was cited.

Comments 3:

Discussions: you have a good introduction, but still there are new citations added in the discussion chapter. Please make sure that you do not have new citations in the discussion section, hence improve the introduction so that you give a better status quo to which you compare your results with.

Response 3: Thank you for your suggestion. We have revised the Introduction and Discussions.

We are grateful for your help.

Reviewer 4 Report

Comments and Suggestions for Authors

The authors have conducted well-designed research with a very important aim. Although they have impressive sample and are focused on the main aim with their instruments and have chosen appropriate analysis, have a balanced introduction, there are also some shortcomings that should be addressed:

Page 2, line 96 – what is meant by natural correlates?

Please comment on the response rate – could students reject participation? Due to their age (I suppose they are 12-13y.o.), did you receive parental consent?

Students’ information would be protected and remain anonymous. “ – I would suggest this be part of the Participants section

„National Taiwan Normal University's Institutional Review Board  approved this study (201905HS068).“ I would suggest this be put in the Ethics section which should be added with information about all the procedures done to conduct research in line with ethical guidelines.

Instruments should be described in a way that it is known not only what were the questions, but also what were the possible answers (for 2.3.1. to 2.3.4.).

For Table 5, I would recommend adding the total variance of the criteria explained. Also, it would be much more convenient for readers if the table and text explaining the results in the table were next to each other.

In line 229, page 5, the authors introduce online information risk. I would suggest using online privacy risk consistently in text.

Another study has shown that Tinder  users don’t just swipe to find potential matches, but also to gather insights on how to 258 present themselves in a way that would appeal to others [45].“  - Authors have not studied this, and I don't see the relevance of that information in the discussion.

Authors say “In the present study, a multivariate analysis model negatively associated higher levels of parental restrictive mediation with cyberbullying victimization,“ – what does that mean? If you say that parental restrictive mediation is negatively associated with victimization, means, in accordance with the instrument description, that those whose parents more restrictively mediate their internet use are less likely to be victimized. But adding higher levels of, it would mean the opposite. And results suggest that those who are lower on restrictive mediation, are more prone to be victimized. In that paragraph, the authors do not discuss the result that higher active mediation is associated with more cyberbullying victimization – you have very interesting data that should be addressed in the discussion. Especially since the authors say “At least one study has shown that adolescents with positive parenting are less likely to be involved in the perpetration of relationship abuse and victimization [60].“ Your data suggest that strict, rather than positive, and active mediation is a protective factor. Is it due to instruments, cultural norms, or something else?

Author Response

Thank you very much for taking the time to review this manuscript. Please find the detailed responses below and the revisions/corrections highlighted/in track changes in the re-submitted files.

Comments 1:

The authors have conducted well-designed research with a very important aim. Although they have impressive sample and are focused on the main aim with their instruments and have chosen appropriate analysis, have a balanced introduction, there are also some shortcomings that should be addressed:

Page 2, line 96 – what is meant by natural correlates?

Response 1: Thank you for your suggestion. We have changed the word to “correlates”.

Comments 2:

Please comment on the response rate – could students reject participation? Due to their age (I suppose they are 12-13y.o.), did you receive parental consent?

Response 2: Thank you for your suggestion. We have revised 2.1 Participants. We sent consent forms to students and their parents. Students could reject participation.

Comments 3:

“Students’ information would be protected and remain anonymous. “ – I would suggest this be part of the Participants section

Response 3: Thank you for your suggestion. We have moved this sentence to 2.1 Participants.

Comments 4:

„National Taiwan Normal University's Institutional Review Board  approved this study (201905HS068).“ I would suggest this be put in the Ethics section which should be added with information about all the procedures done to conduct research in line with ethical guidelines.

Response 4: Thank you for your suggestion. We have added information about the ethics of this study in 2.2 Instrument.

Comments 5:

Instruments should be described in a way that it is known not only what were the questions, but also what were the possible answers (for 2.3.1. to 2.3.4.).

Response 5: Thank you for your suggestion. We have added the possible response in 2.3.1 to 2.3.4.

Comments 6:

For Table 5, I would recommend adding the total variance of the criteria explained. Also, it would be much more convenient for readers if the table and text explaining the results in the table were next to each other.

Response 6: Thank you for your suggestion. We have added the total variance of the criteria explained in the footnote of Table 5. The tables’ locations in the final version will be arranged by the editors.

Comments 7:

In line 229, page 5, the authors introduce online information risk. I would suggest using online privacy risk consistently in text.

Response 7: Thank you for your suggestion. We have revised it.

Comments 8:

“Another study has shown that Tinder  users don’t just swipe to find potential matches, but also to gather insights on how to 258 present themselves in a way that would appeal to others [45].“  - Authors have not studied this, and I don't see the relevance of that information in the discussion.

Response 8: Thank you for your suggestion. We have removed this sentence.

Comments 9:

Authors say “In the present study, a multivariate analysis model negatively associated higher levels of parental restrictive mediation with cyberbullying victimization,“ – what does that mean? If you say that parental restrictive mediation is negatively associated with victimization, means, in accordance with the instrument description, that those whose parents more restrictively mediate their internet use are less likely to be victimized. But adding higher levels of, it would mean the opposite. And results suggest that those who are lower on restrictive mediation, are more prone to be victimized. In that paragraph, the authors do not discuss the result that higher active mediation is associated with more cyberbullying victimization – you have very interesting data that should be addressed in the discussion. Especially since the authors say “At least one study has shown that adolescents with positive parenting are less likely to be involved in the perpetration of relationship abuse and victimization [60].“ Your data suggest that strict, rather than positive, and active mediation is a protective factor. Is it due to instruments, cultural norms, or something else?

Response 9: Thank you for your suggestion. We have added the discussion of parental mediation.

We are grateful for your help.

Round 2

Reviewer 2 Report

Comments and Suggestions for Authors

I have made note that the suggested revisions regarding the sample and methodology of the study have been made in the second version. However, the biological sex can still be a limiting factor for adolescent use in behavioral studies. Thus, I would highly recommend that you regard the gender identity with more flexibility in your future studies. With these updates, the article can be considered for publishing in the related issue.  

Comments on the Quality of English Language

Proofreading essential, but no major problems.